# Haspin kinase modulates nuclear architecture and Polycomb-dependent gene silencing

Ujué Fresán[1,2©], Maria A. Rodríguez-Sánchez[1©], Oscar Reina[3], Victor G. Corces[4], M. Lluisa Espinàs[1,2]*

1 Institut de Biologia Molecular de Barcelona, IBMB-CSIC, Barcelona, Spain, 2 Institute for Research in Biomedicine IRB, Barcelona, Spain, 3 Bioinformatics and Biostatistics Unit, Institute for Research in Biomedicine IRB, Barcelona, Spain, 4 Department of Human Genetics, Emory University School of Medicine, Atlanta, Georgia, United States of America

© These authors contributed equally to this work.
* mlebmc@ibmb.csic.es

**Data Availability Statement:** All relevant data are within the manuscript and its Supporting Information files except ChIP-seq files that are

## Abstract

Haspin, a highly conserved kinase in eukaryotes, has been shown to be responsible for phosphorylation of histone H3 at threonine 3 (H3T3ph) during mitosis, in mammals and yeast. Here we report that haspin is the kinase that phosphorylates H3T3 in *Drosophila melanogaster* and it is involved in sister chromatid cohesion during mitosis. Our data reveal that haspin also phosphorylates H3T3 in interphase. H3T3ph localizes in broad silenced domains at heterochromatin and lamin-enriched euchromatic regions. Loss of haspin compromises insulator activity in enhancer-blocking assays and triggers a decrease in nuclear size that is accompanied by changes in nuclear envelope morphology. We show that haspin is a suppressor of position-effect variegation involved in heterochromatin organization. Our results also demonstrate that haspin is necessary for pairing-sensitive silencing and it is required for robust Polycomb-dependent homeotic gene silencing. Haspin associates with the cohesin complex in interphase, mediates Pds5 binding to chromatin and cooperates with Pds5-cohesin to modify Polycomb-dependent homeotic transformations. Therefore, this study uncovers an unanticipated role for haspin kinase in genome organization of interphase cells and demonstrates that haspin is required for homeotic gene regulation.

## Author summary

Haspin is a highly conserved kinase in eukaryotes involved in chromosome organization during mitosis. In this study we demonstrated that haspin is also required to maintain proper chromatin organization during interphase. Our analyses showed that *Drosophila* haspin is necessary for insulator activity, nuclear architecture, heterochromatin organization and pairing-sensitive gene silencing. We further found that haspin modulates Pds5-cohesin association with chromatin and it is required for robust Polycomb-mediated homeotic gene silencing. Overall our findings reveal that haspin kinase is a key element in chromatin organization and thereby regulates gene expression.

available from the NCBI GEO database (accession number GSE98223).

**Funding:** This work was supported by the Spanish Ministerio de Economia y Competitividad (BFU2013-48712-P) and the Generalitat de Catalunya (SGR2017-475). U. F. and M. R-S. acknowledge receipt of doctoral fellowships from Consejo Superior de Investigaciones Cientificas and Ministerio de Economia y Competitividad respectively. The funders had no role in study design, data collection and analysis, decision to publish, or preparation of the manuscript.

**Competing interests:** The authors have declared that no competing interests exist.

# Introduction

Genome organization in the cell nucleus plays an important role in the regulation of gene expression during cellular differentiation and development [1,2]. Insulator or architectural proteins are essential components of the three-dimensional organization of chromatin by mediating long-range interactions between distant sites in the genome. Current results suggest that architectural complexes have two inter-related functions: to organize the genome in domains and to facilitate the interaction between regulatory elements [3,4,5]. Several architectural proteins have been characterized in *Drosophila melanogaster*, including DNA-binding proteins (CTCF, SuHw, BEAF-32, GAGA, DREF, TFIIIC, Z4, Elba, ZIPIC, Ibf1 and Ibf2) that recruit accessory factors (CP190, mod(mdg4), Rad21, Cap-H2, Fs(1)h-L, L(3)mbt and chromator) to mediate chromatin interactions [6]. A small number of architectural proteins have been characterized in mammals, and among them, CTCF and cohesin, have also been shown to support interactions between distant sites in the genome [5].

Long-range gene regulation also involves epigenetic components such as the Polycomb group proteins (PcG) [7,8,9,10,11]. In *Drosophila*, homeotic (Hox) genes, which encode evolutionary conserved master regulators of development, are the most prominent PcG targets. Precise spatiotemporal expression of Hox genes involves an intricate collection of enhancers, promoters, polycomb response elements (PREs) and insulators. It has been demonstrated, both by fluorescent *in situ* hybridization and Chromosome Conformation Capture approaches, that chromatin organization of the *Abdominal-B* (*Abd-B*) locus in the bithorax complex (BX-C) is a critical determinant of the regulation of the expression of the gene. Several reports have shown that insulators and PREs interact with the *Abd-B* promoter in tissues where the gene is not expressed, and Polycomb and the insulator/architectural proteins CTCF and CP190 are required for these interactions [12,13,14,15].

Histone modifications and the enzymes responsible for them are also important players in the regulation of chromatin organization. Haspin, a highly conserved kinase in eukaryotes, is responsible for phosphorylation of histone H3T3 during mitosis [16]. Haspin kinase has been shown to be involved in sister chromatid cohesion [17] and to be necessary to localize the Chromosomal Passenger Complex (CPC) on mitotic chromatin at centromeres to activate Aurora B that regulates kinetochore-microtubule attachments [18,19,20]. In fission yeast and mammalian cells, haspin has been shown to bind the cohesin-associated protein Pds5 at centromeres and to antagonize the cohesin-unloading factor Wapl [21,22,23]. H3T3ph dephosphorylation upon exit from M phase has been shown to be necessary for chromosome decondensation and nuclear envelope reformation [24]. Haspin kinase has been reported to be strongly activated by Cdk1 and Polo-like kinase in mitosis [25,26]. However, haspin contains an atypical protein kinase domain, which is conserved from yeast to humans, that does not require phosphorylation on the activation loop for activity, suggesting that it could be partially active all along the cell cycle. We show here that haspin is necessary for insulator activity, position-effect variegation (PEV) and pairing-sensitive silencing (PSS) modulating nuclear architecture in interphase. We also demonstrate that haspin is required for robust Polycomb-dependent homeotic gene silencing. Altogether our results indicate that *Drosophila* haspin kinase is involved, not only in chromosome organization during mitosis, but also in genome organization in interphase cells playing a functional role in gene regulation.

# Results

## Haspin kinase is necessary for insulator activity

In order to identify new proteins with insulator activity we performed a mutagenesis screen in *Drosophila* by random transposition of a P element. New insertions were analyzed for changes

in reporter gene expression in enhancer-blocking assays using a transgenic line that contain the *Fab7* boundary/insulator element of the BX-C between the *white* enhancer and the *mini-white* reporter gene (B7[27.1], S1 Fig), which blocks promoter activation by the distal enhancer. Any relief of insulator activity should allow communication between enhancer and promoter of the *white* reporter gene increasing eye color. Among all the lines showing a significant relief of enhancer-blocking we further studied line 86 (Fig 1A). The P element insertion in this line was mapped to the first exon in the 5'UTR of the gene *haspin* (S2A Fig). We analyzed *haspin* expression in the homozygous line, and we observed a strong reduction in transcript levels indicating that *haspin*[86] is either a null or a strong hypomorph *haspin* alelle (S2B Fig). By mobilazing the P element in line 86 we obtained line 128 that harbors a partial deletion of the P element, the first and second exons and part of the second intron of the gene *haspin* (S2A Fig), likely rendering the gene non functional. *Haspin*[128] homozygous mutant flies are viable, they show a decrease in adult longevity which is stronger in females than males (S2C Fig), and fertility of both sexes is clearly affected (S2D Fig).

To establish further the role of haspin in insulator function we performed enhancer-blocking assays with independent mutant backgrounds and different insulator elements. We knocked down haspin levels using the UAS/Gal4 system. Our results show a clear increase in the eye pigmentation of flies with decreased haspin levels using another transgenic line that contains the *Fab7* element of the BX-C (Fig 1B). Indeed, quantitative analyses showed significant relief of *Fab7* enhancer-blocking activity in the different haspin mutant lines, which is similar to the one observed in CP190 insulator protein mutant background (Fig 1C). Moreover, similar effects were obtained using transgenic lines containing either the *Fab6* or *Fab8* insulator elements between the *white* enhancer and the *mini-white* reporter gene (Fig 1D). These results indicate that haspin is necessary for the function of different insulators of the bithorax complex and suggest that haspin could be involved in modulating higher-order chromatin organization of the BX-C.

## *Drosophila* haspin is the kinase responsible for the phosphorylation of histone H3 at Thr3 in mitosis and interphase

Haspin is a highly conserved kinase that has been shown to be responsible for mitotic phosphorylation of histone H3 at threonine 3 at centromeric regions in yeast and mammals [27]. To analyze whether haspin is also performing this function in *Drosophila* we carried out immunostaining assays of spread metaphase chromosomes of *Drosophila* larval brains using antibodies against H3T3ph and the centromere marker Cenp-C. Our results show that H3T3ph is concentrated at the inner centromere between the paired regions of Cenp-C and that H3T3ph signal completely disappears in a *haspin*[128] mutant background (Fig 2A and S3A Fig). These assays were performed in the presence of colcemid which arrest cells in a prolonged prometaphase that leads to the opening of sister chromatid arms to produce X-shaped chromosomes [28]. Our assays show that haspin depletion disrupts connection between sister chromatids at the centromeres (S3A Fig, see differences in connection between chromatids in DNA panels), as previously reported in mammals [17]. Indeed, the inter-kinetochore distance on chromosome spreads prepared from *haspin*[128] mutant brains, which was measured using the centromere marker Cenp-C, was 30% further apart compared to control (S3B Fig). Thus, the mitotic functions of haspin are likely to be conserved in *Drosophila*.

Most mitotic kinases need to be phosphorylated at the activation loop to be activated and this only occurs during mitosis [29]. Although haspin kinase has also been shown to be strongly activated in mitosis by phosphorylation, it is intrinsically active without phosphorylation, suggesting that it could be partially active all along the cell cycle [27]. Therefore, in order

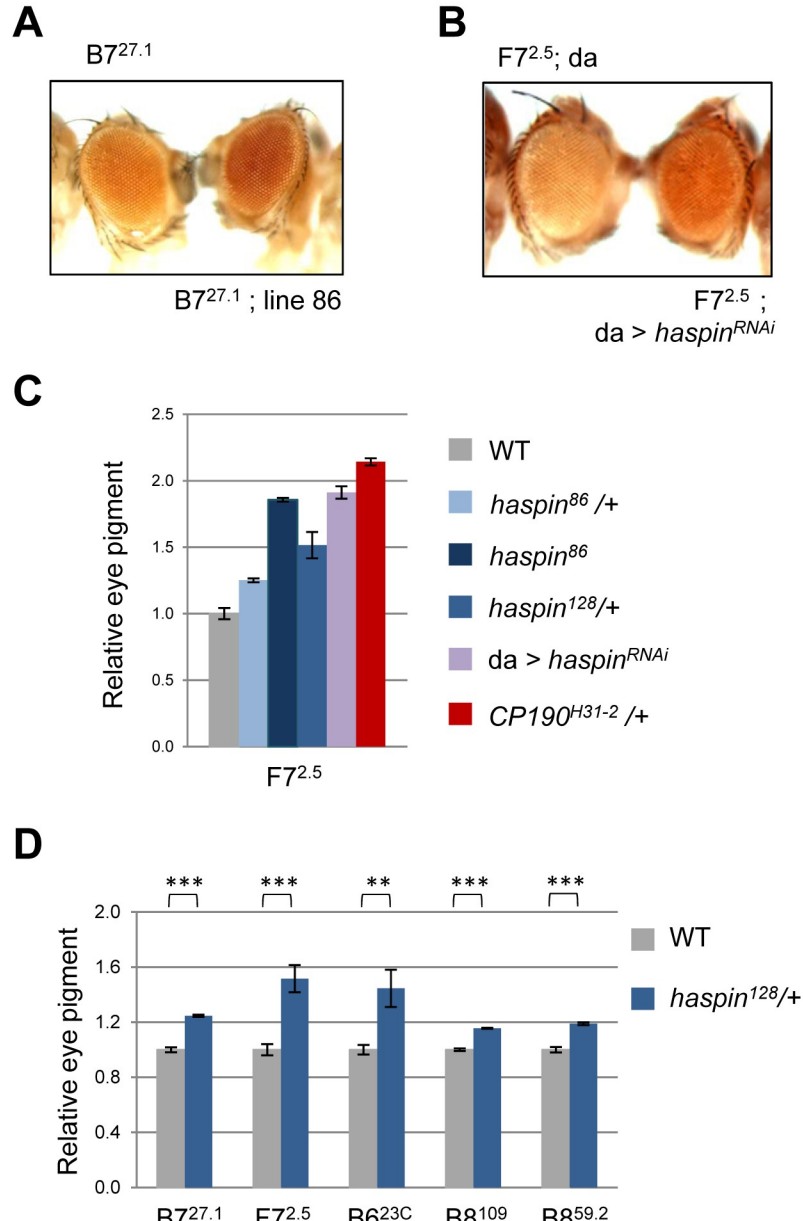

**Fig 1. Insulator activity of haspin.** A) Eye color of representative flies of enhancer-blocking assays using the transgenic line B7[27.1] that contains the *Fab7* insulator element in wild-type and heterozygous line 86 background. B) Eye color of representative flies of enhancer-blocking assays using the transgenic line F7[2.5] that contains the *Fab7* insulator and PRE elements in haspin RNAi mutant background. C) Quantitative analysis of eye pigment in flies with the F7[2.5] construct and the genotypes indicated. n = 3, significant differences between wild-type and the different mutant backgrounds as determined by Student's *t*-test (p<0.001). D) Quantitative analysis of eye pigment in flies with constructs containing *Fab6*, *Fab7* and *Fab8* insulator elements (see S1 Fig for details of the constructs) in wild-type and *haspin*[128] heterozygous mutant background (n = 3). Statistical significance (**p<0.01 and ***p<0.001) was determined by Student's *t*-test.

to characterize whether H3 is phosphorylated at Thr3 during interphase we performed immunostaining assays of polytene chromosomes of *Drosophila* salivary glands. As shown in Fig 2B, H3T3ph signals are observed at discrete loci on extended polytene chromosomes in wild-type

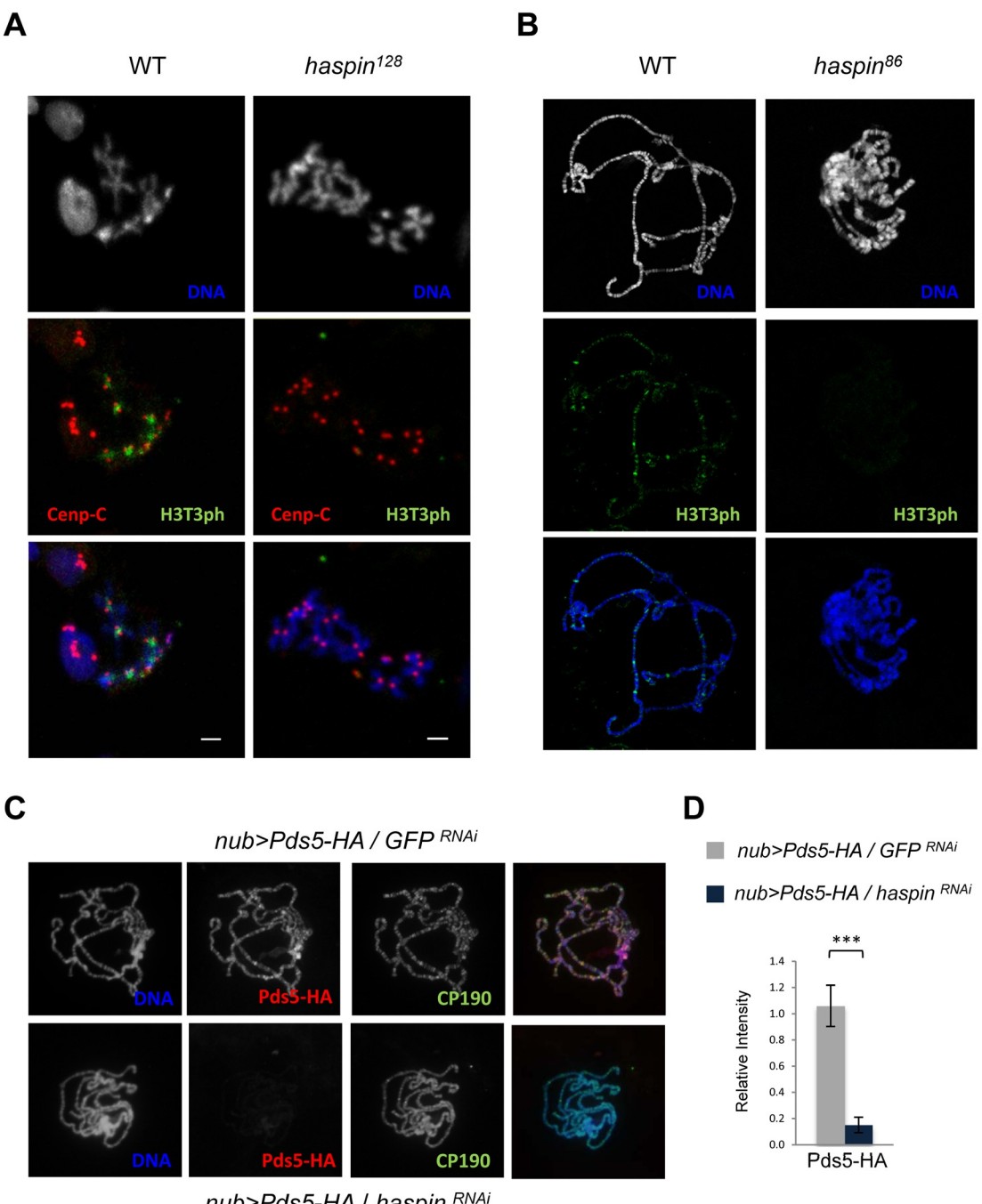

**Fig 2. Haspin kinase phosphorylates H3T3 in mitosis and interphase and it is required for chromatin binding of Pds5.** A) Immunolocalization of H3T3ph (green) and Cenp-C (red) in mitotic cells of wild-type and *haspin*[128] mutant *Drosophila* larval brains. DNA is stained with DAPI (blue). Scale bars are 2μm. B) Representative polytene chromosome spreads from wild-type and *haspin*[86] mutant salivary glands of third-instar larvae immunostained with antibodies against H3T3ph (green). DNA is stained with DAPI (blue). C) Representative polytene chromosome spreads from salivary glands of third-instar larvae that express Pds5-HA under the control of *nubbin* promoter in GFP (upper panels) or haspin (lower panels) RNAi backgrounds, immunostained with antibodies against HA (red) and CP190 (green). DNA is stained with DAPI (blue). D) Pds5-HA/CP190 immunofluorescence intensity ratio in polytene chromosome spreads from salivary glands of third-instar larvae that express Pds5-HA under the control of *nubbin* promoter in GFP or haspin RNAi backgrounds. n = 5, means and s.d. are shown. ***p<0.001 was determined by Student's *t*-test.

larvae while these signals are not present in a mutant *haspin[86]* background (Fig 2B), indicating that haspin is required for the phosphorylation of histone H3 at Thr3 in interphase.

## Chromatin binding of cohesin-associated protein Pds5 depends on haspin

Haspin and the cohesin complex have been shown to be functionally related during mitosis in mammals and yeast. Haspin is involved in protection of mitotic centromere cohesion by binding to the cohesin-associated protein Pds5 and antagonizing the cohesin unloading factor Wapl [21,22,23]. Since our results show that haspin is required for centromeric cohesion in *Drosophila* (S3 Fig), we asked whether the interaction of haspin with Pds5 is also conserved and, moreover, if this association takes place not only during mitosis but also in interphase. To characterize the interaction between haspin and Pds5 during interphase we performed coimmunoprecipitation experiments in larval salivary gland nuclear extracts of a transgenic line that express Pds5-HA-Flag. To this end, we raised antibodies against haspin (see Materials and Methods and S4A Fig) that were able to specifically coprecipitate Pds5-HA-Flag (S4B Fig). To further characterize this interaction we performed Pds5 immunolocalization assays in Pds5-HA-Flag larval salivary gland polytene chromosomes in either mock (GFP) or haspin RNAi backgrounds. Our results show a strong decrease of Pds5 binding to chromosomes when knocking-down haspin levels (Fig 2C and 2D). Since the lack of Pds5 binding to chromatin could also reflect a contribution of haspin to synthesis and/or stability of the protein we have analyzed Pds5 mRNA and protein levels. While there are no significant changes in Pds5 mRNA levels (S4C Fig), protein levels are clearly reduced in larval salivary glands (S4D Fig). These results strongly suggest destabilization of Pds5 protein due, most likely, to its inability to associate with haspin and chromatin. Altogether, our results indicate that haspin interacts with the cohesin-associated protein Pds5 during interphase and it is required for Pds5 binding to chromatin.

It has been shown that Pds5 proteins have both positive and negative effects on cohesin association with chromatin; they cooperate with Wapl in releasing cohesin from DNA but they have also been implicated on cohesion during mitosis [30]. To characterize the relationship between haspin and the core of the cohesin complex we performed coimmunoprecipitation experiments using salivary gland nuclear extracts that ubiquitously express the cohesin subunit Rad21 fused to a myc epitope in a Rad21 mutant background (*Vtd[ex3]*). αhaspin antibodies were able to specifically coprecipitate Rad21-myc (S5A Fig) indicating that haspin associates with the core of the cohesin complex during interphase. We then performed immunolocalization assays in larval salivary gland polytene chromosomes that showed no apparent changes in Rad21 binding to chromosomes in the absence of haspin (S5B Fig, compare upper and lower panels). We also analyzed Rad21 expression and our results showed that transcript levels are similar in haspin mutant background compared to control (S5C Fig). To further analyze whether haspin modulates the interaction of this complex with chromatin we quantified the amount of chromatin-associated cohesin in the absence of haspin. To this end, formaldehyde crosslinking of *Drosophila* embryo nuclei that express Rad21-myc in a Rad21 mutant background (*Vtd[ex3]*) in either control or *haspin[128]* mutant background were carried out, chromatin fractions were purified and the amount of Rad21 and Histone H3 were compared. During the first hours of *Drosophila* embryo development nuclei go through 13 rapid mitotic divisions having very short S-phases and omitting gap phases while later in development the mitotic rate is much slower. Therefore, in early embryos cells are most of the time in the mitotic phase of the cell cycle while in late embryos most of cells are in interphase. Our assays showed a significant decrease in chromatin-bound Rad21 in the absence haspin in early embryos (0–4 hours) that, although attenuated, is also present in late embryos (20–24 hours)

(S5D and S5E Fig). Altogether, these data indicate that haspin modulates the dynamic association of cohesin with chromatin during cell cycle.

## Nuclear morphology defects and nuclear compaction during interphase in the absence of haspin activity

We performed immunostaining experiments in wild-type and haspin-mutant *Drosophila* late third-instar larval salivary glands. We found that haspin-depleted cells displayed irregularly shaped nuclei showing a crumpled raisin-like appearance revealed by lamin Dm0 immunolocalization (Fig 3A). To characterize the subcellular localization of haspin we performed biochemical fractionation of *Drosophila* embryos that express an epitope-tagged haspin-HA protein. A significant amount of haspin-HA was detected in the nuclear matrix fraction, which is characterized by the presence of lamin Dm0 (S6A Fig). Insulator protein CP190 and Polycomb group of proteins were also found mostly associated with the nuclear matrix (S6A Fig), consistent with previously reported results [31,32]. These data indicate that haspin localizes at the nuclear lamina and modulates nuclear morphology.

In the assays reported above, changes in nuclear size between control and haspin-depleted salivary glands are apparent (Fig 3A). We further characterized this phenotype in detail by DAPI immunostaining experiments in either loss or gain of function alleles of haspin. Our analyses showed that in late third-instar larvae, at a time when there is no further replication, haspin mutants exhibit reduced nuclear size (S6B Fig, compare WT and *haspin*[86] panels). Quantification analyses confirmed the decrease in the absence of haspin (Fig 3B). On the contrary, overexpression of haspin in a transgenic line with two extra copies of the *haspin* genomic locus gave rise to a clear increase in the nuclear size of larval salivary glands (Fig 3B and S6B Fig). The expression of haspin under the control of its own promoter in a mutant background totally rescued the loss of function phenotype (Fig 3B and S6B Fig, no significant differences in nuclear size were observed between WT and *haspin*[86]; *haspin*[PROM]). These altered nuclear sizes are not the consequence of changes in DNA replication since both WT and haspin mutant salivary glands showed similar DNA content (S6C Fig). When we knocked-down haspin levels in salivary gland nuclei using the UAS/Gal4 system and the *nubbin-Gal4* driver we also observed a strong decrease in nuclear size (Fig 3B, nub > *haspin*[RNAi] lane). Similar effects were obtained by knocking-down two well characterized insulator/architectural proteins such as CP190 and CTCF (Fig 3B). In order to characterize the contribution of the kinase domain of haspin to this phenotype we obtained transgenic lines that express epitope-tagged haspin-HA proteins either wild-type or with a Lys282Met substitution in the ATP recognition motif or a His420Ala, change that has been previously shown to be required for haspin activity in mammals [33]. Similar levels of expression of the epitope-tagged proteins in the different transgenic lines were obtained using the UAS/Gal4 system (S6D Fig). Our analyses showed that while overexpression of haspin-HA wild-type protein in a null mutant background was able to totally rescue the nuclear size of salivary gland cells, overexpression of kinase dead proteins were not (Fig 3C). Thus, haspin kinase activity is required for global nuclear organization in interphase cells.

Altogether our results show extensive interphase nuclear compaction in haspin-depleted cells that is associated with defects in nuclear morphology and they suggest that haspin kinase is an enzymatic protein that associates with the nuclear matrix/lamina to perform key roles in chromatin organization and nuclear architecture.

## Haspin is a suppressor of position-effect variegation

In order to identify genome-wide location of H3T3ph we performed chromatin immunoprecipitation sequencing (ChIP-seq) analysis in S2 cells using αH3T3ph antibody. Sites of

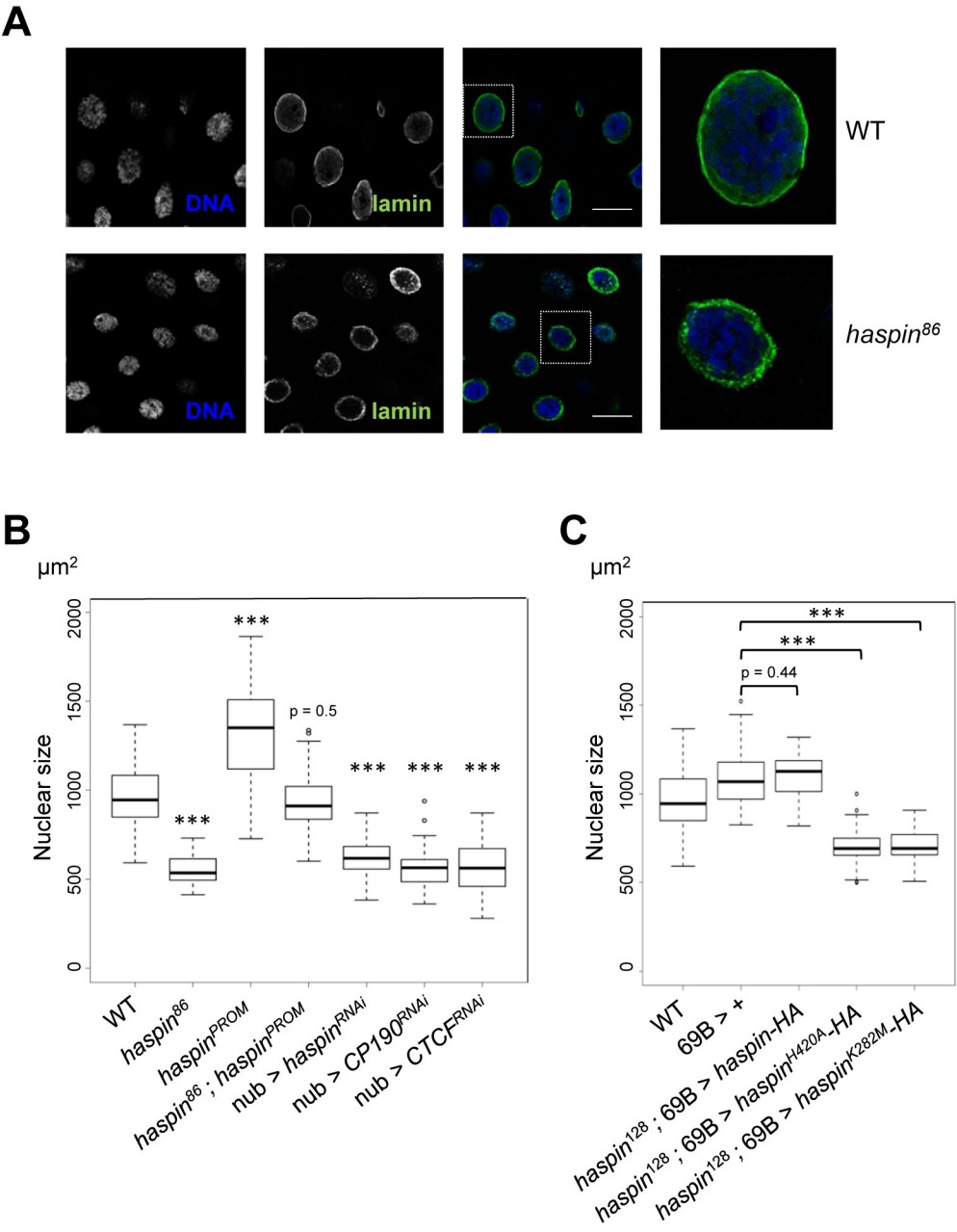

**Fig 3. Modulation of nuclear architecture by haspin kinase.** A) Immunostaining of *Drosophila* salivary glands with antibodies against lamin Dm0 in wild-type third-instar larvae (upper panel) and *haspin*[86] mutant background (lower panel). DNA is stained with DAPI. In the overlap (right panels), lamin Dm0 is shown in green and DAPI in blue. Scale bars are 50 μm. B) Box plot showing quantification of DAPI signals of *Drosophila* salivary gland nuclei in wild-type (WT), haspin mutant background (*haspin*[86]), overexpression of haspin under control of its own promoter (*haspin*[PROM]), overexpression of haspin in a mutant background (*haspin*[86]; *haspin*[PROM]) and in haspin, CP190, and CTCF RNAis under the control of *nubbin* promoter (nub > *haspin*[RNAi], nub > *CP190*[RNAi], and nub > *CTCF*[RNAi]). Nuclear size was determined in around 50 nuclei for each condition (n≥3). The p values as determined by Wilcoxon test of the different genetic backgrounds respect to WT are indicated (*** p < 0.001). C) Box plot showing quantification of DAPI signal of *Drosophila* salivary gland nuclei in wild-type (WT) and overexpression under the control of 69B of tagged versions of either haspin wild-type (*haspin-HA*) or haspin with a mutated kinase domain (*haspin*[H420A]*-HA* and *haspin*[K282M]*-HA*). Nuclear size was determined in around 50 nuclei for each condition (n≥3). Significant differences between wild-type and mutated kinase domains as determined by Wilcoxon test (*** p<0.001).

enrichment (peaks) were identified by bioinformatics and biostatistics analyses of aligned ChIP-seq data (see experimental procedures). H3T3ph enriched regions across whole genome accumulate in heterochromatic regions of the chromosomes where they colocalize with HP1a (Fig 4A). To statistically assess the overlap between H3T3ph and HP1a binding sites we used overlap permutation tests that showed a high degree of statistically significant association (z-score >38, p-value < 0.01 for all HP1a modENCODE replicates, S7A Fig). We also examined the location of H3T3ph peaks in relation to the *Drosophila melanogaster* 9 different chromatin states as determined in S2 cells [34] and we found that they are preferentially located in state 7, which correspond to centromeric heterochromatin and chromosome 4 (S7B Fig). Indeed, strong association between H3T3ph and heterochromatin regions was statistically assessed using overlap permutation tests (positive z-scores for states 7 and 8, S7B Fig). The model generated by Kharchenko et al. [34] distinguishes a set of heterochromatin-like regions (state 8) enriched in chromatin marks typically associated with heterochromatic regions (H3K9me2/me3 and HP1a) but at lower levels than in pericentromeric heterochromatin. This state occupies extensive domains in euchromatic arms of the chromosomes such as the region shown in Fig 4B. Line on top corresponds to the H3T3ph binding profile (normalized H3T3ph IP versus input signal) while line on bottom corresponds to H3T3ph enriched peaks. Our assays showed that H3T3ph is enriched at these heterochromatin-like regions (Fig 4B and S7B Fig).

Our ChIP-seq analyses do not allow distinguishing if the heterochromatic H3T3ph enrichment is only present in mitotic cells or it is also found in interphase cells, since we have shown above that H3T3ph signal is strongly activated during mitosis in centromeric regions in *Drosophila* (Fig 2A) as it has been previously reported in mammals [25,26]. To study whether haspin could be involved in heterochromatin organization in interphase we analyzed position-effect variegation (PEV) in a haspin mutant background. In the *In(1)w^{m4}* rearrangement the inversion results in juxtaposition of the *white* gene with the heterochromatic region of the X chromosome. In this line spreading of heterochromatin into the euchromatic domain results in silencing and variegation of *white* expression [35]. Our analyses showed that *haspin^{128}* heterozygous mutation strongly suppressed *white* silencing in the *In(1)w^{m4}* rearrangement (Fig 4C), indicating that *haspin* is a novel *Su(var)* gene (suppressor of PEV), such as *Su(var)2-5* (HP1a) and *Su(var)3-9* (histone H3K9 methyltransferase) [35]. Thus, our results indicate that haspin is involved in heterochromatin organization and suggest that H3T3 phosphorylation by haspin modulates heterochromatin formation and/or stability.

Our assays showed that H3T3ph enriched regions found at euchromatic arms of the chromosomes are often localized in broadly enriched large genomic regions usually flanked by insulator proteins such as CP190 and Ibf2. While some of these regions are enriched in HP1a, such as region shown in Fig 4B, others were found to be enriched in lamin Dm0. An example of these regions is shown in Fig 4D. We also used overlap permutation tests to statistically assess the location of H3T3ph peaks at euchromatic arms of the chromosomes in relation to the different chromatin states [34]. H3T3ph euchromatic regions were defined as those not overlapping with chromatin state 7 and we found that they are preferentially located in state 9, which corresponds to silent domains (S7C Fig). H3T3ph enriched regions were often found in clusters presenting a high number of regions in very close succession while other areas with moderate H3T3ph signal presented a lower number of sparsely reported regions, as it is shown in line at the bottom in Fig 4B and 4D and S8A Fig. Depending on the density of accumulation, H3T3ph enriched regions were described as highly or lowly enriched by visual inspection (S8A Fig, regions b and c respectively). We observed that while high (dense) H3T3ph enriched regions tended to colocalize with lamin Dm0, low (sparse) enriched regions colocalized with H3K27me3, such as region c that corresponds to the Antennapedia complex, suggesting that PcG silenced domains contain low levels of H3T3ph. Our ChIP-seq data also suggest that

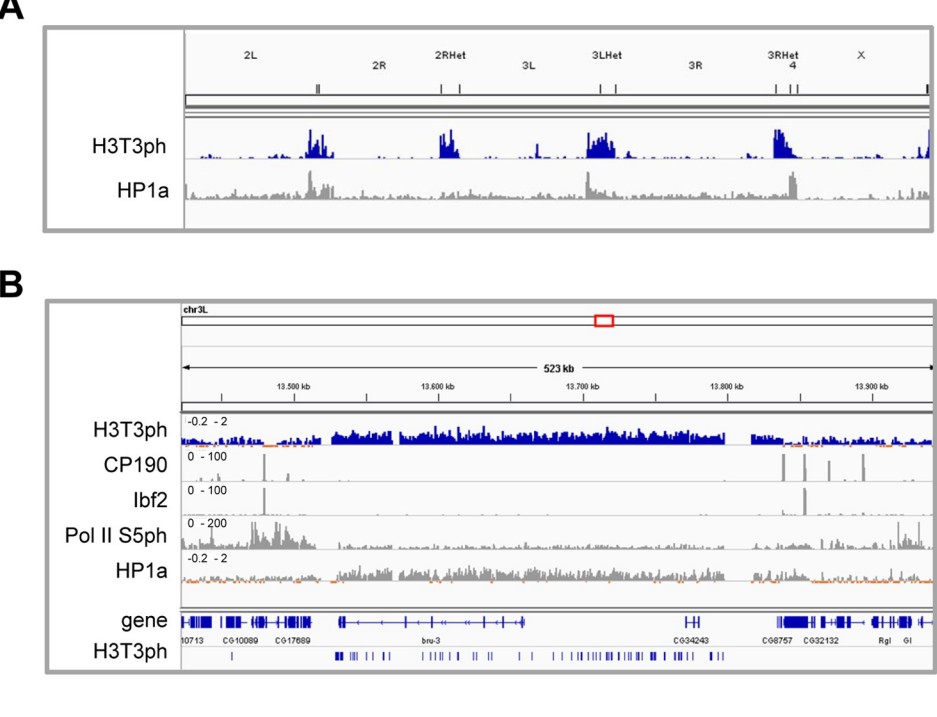

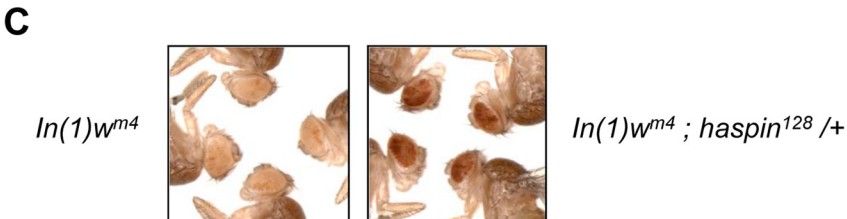

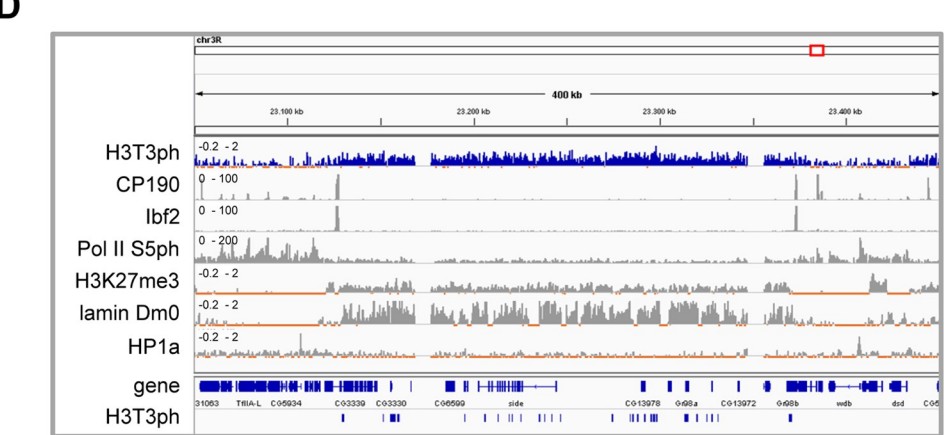

**Fig 4. Haspin is a suppressor of position-effect variegation.** A) ChIP-seq data for H3T3ph genome-wide localization. Enriched regions of HP1a (modENCODE data, see S1 Text) are depicted below. B) ChIP-seq data for H3T3ph over a region of 500 kb in chromosome 3 in *Drosophila* S2 cells. Binding profiles of CP190 and Ibf2 [66], Pol II S5ph and HP1a (modENCODE data) are shown. H3T3ph peaks are depicted in last row. C) Effect of *haspin*[128] mutation on position-effect variegation in *In(1)w*[m4] males (n = 3). D) Representative ChIP-seq data for H3T3ph over a 400 kb region of chromosome 3R in *Drosophila* S2 cells. Binding profiles of CP190 and Ibf2 [66], Pol II S5ph, H3K27me3 and HP1a (modENCODE data, see S1 Text) and lamin Dm0 [69] are shown. H3T3ph peaks are depicted in last row.

H3T3ph enriched regions do not contain binding sites for the active form of the RNA polymerase II phosphorylated at serine 5 (Fig 4B and 4D and S8A Fig). Indeed, general comparisons of H3T3ph with active RNA polymerase II binding sites showed a mutually antagonistic location (S8B Fig). Altogether, our results indicate that phosphorylated histone H3 at Thr3 colocalizes with silent chromatin.

## Haspin is required for robust Polycomb-dependent homeotic gene silencing

Insulator/architectural elements of the BX-C have been shown to be involved in the regulation of homeotic gene expression [36,37,38] and our results reported above indicate that haspin is required for the function of several of these regulatory elements (Fig 1). Therefore, we analyzed levels of homeotic gene expression by reverse transcription followed by quantitative PCR and we found that *Abd-B* gene was derepressed in the mutant (Fig 5A), indicating that haspin is required for *Abd-B* gene silencing. Homeotic gene silencing depends on PcG complexes and we asked whether chromatin association of PcG complexes is affected by haspin. We performed ChIP assays in wild-type and *haspin^128* mutant backgrounds and we analyzed Polycomb binding at several PRE-containing regulatory elements of *Abd-B*. Our results show a significant reduction in Pc binding at *Mcp*, *Fab7*, *Fab8* and the promoter regions A and B of the *Abd-B* gene in the absence of haspin (Fig 5B).

To further characterize the participation of haspin in Polycomb-dependent homeotic gene silencing we looked for phenotypic changes in a sensitized background. We analyzed homeotic transformations already observed in flies with defective Pc silencing, such as the appearance of sex combs on the second (L2-L1) and third (L3-L1) pairs of legs of male flies and a defective wing development indicative of a wing to haltere (W-H) transformation (Fig 5C). We found that in three different *Pc* mutant backgrounds, *Pc^1*, *Pc^3* and *Pc^15* [39,40,41], the frequency of transformation was much higher in flies homozygous for *haspin^128* mutation and heterozygous for the *Pc* mutation compared with animals carrying only the corresponding *Pc* mutation (Fig 5D). Moreover, the frequencies of L2-L1 and W-H transformations were also higher when knocking haspin levels in a *Pc^3* heterozygous background using the UAS/Gal4 system to express *haspin^RNAi* under the control of the *Act5C* promoter (S9A Fig). Therefore, haspin mutation acts as an enhancer of *Pc* mutations and altogether these results indicate that haspin is required for efficient Polycomb-dependent homeotic gene silencing.

PcG complexes are known to mediate silencing not only in cis but also in trans; accordingly, in transgenic lines the PRE-mediated silencing is pairing-sensitive (PSS) being stronger in homozygous than heterozygous individuals [42]. We analyzed PSS in a transgenic line carrying a construct containing the PRE of the *Fab7* element and the *mini-white* reporter gene [43] in *haspin^128* mutant background. Our assays showed a clear increase in eye pigmentation in the absence of haspin (Fig 5E) indicating that PSS depends on haspin. It was shown earlier that, in another transgenic line, insertion of a *Fab7* transgene in the X chromosome 9.6 kb upstream of the *scalloped* (*sd*) gene induces a mutant phenotype, resulting in disruption of wing morphology (transgenic line 5F24 25,2 [44]). This phenotype is pairing-dependent, as it is observed with strong penetrance of up to 85–95% in homozygous females, whereas it is absent in heterozygous females or hemizygous males, and it is attenuated by mutations in PcG genes [44,45]. The *sd* phenotype of homozygous females is strongly suppressed in a *haspin^128* mutant background with 66% of the individuals showing no sd phenotype at all (S9B Fig). Thus, these results indicate that haspin is necessary for PRE pairing-sensitive silencing and suggest that depletion of haspin might enhance homeotic transformations via loss of PRE contacts and progressive decrease in chromatin silencing efficiency through subsequent generations.

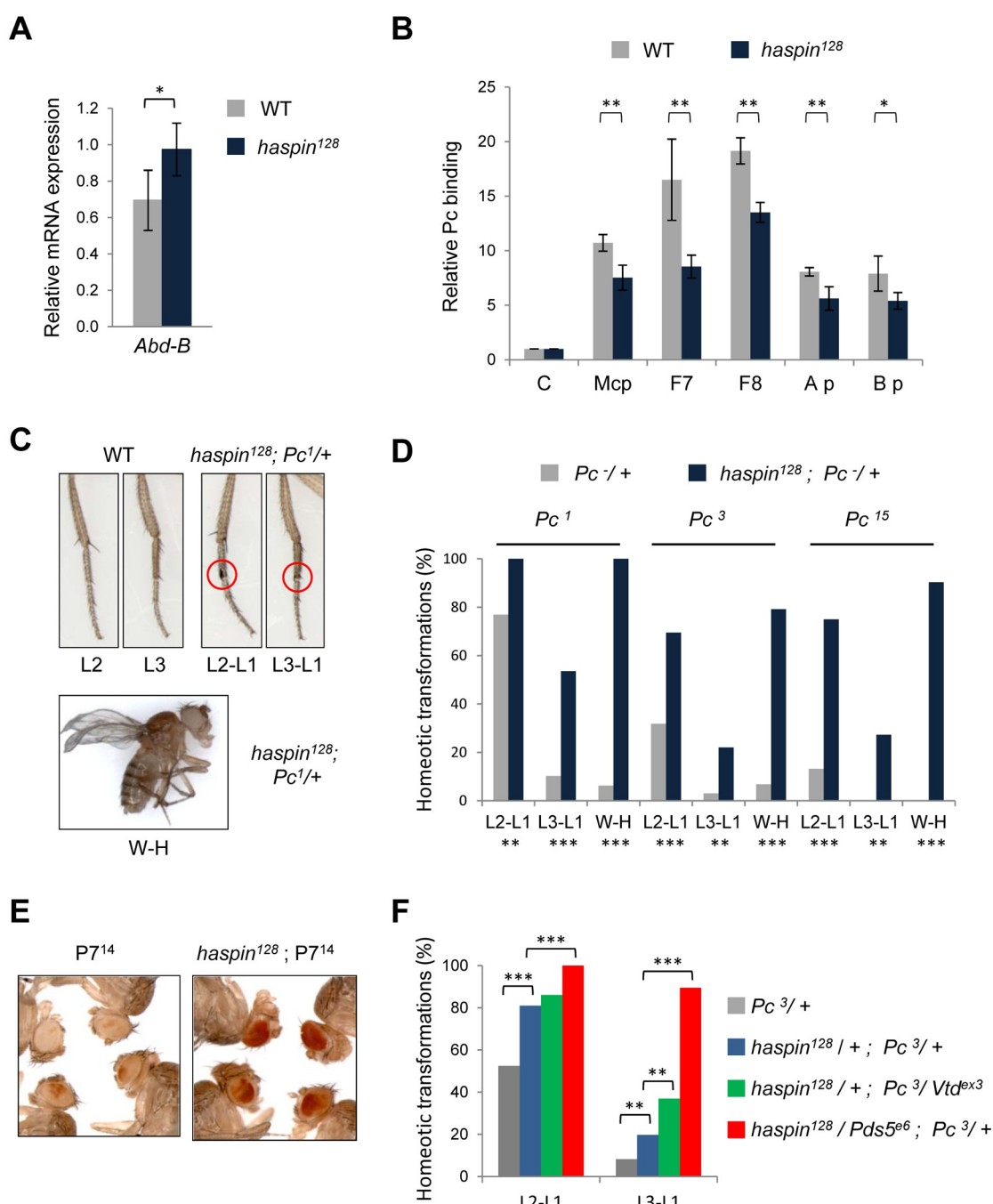

**Fig 5. Haspin is required for Polycomb-dependent homeotic gene silencing.** A) *Abdominal-B* transcriptional levels normalized to *Actin5C* in wild-type or *haspin*[128] mutant *Drosophila* larval brains. n = 4, means and s.d. are shown. B) ChIP-qPCR using an antibody against Polycomb and primers for pointed as a control negative region, *Mcp*, *Fab7* and *Fab8* regulatory elements and *Abd-B* promoters A and B. Average enrichments (normalized to the input sample) are plotted as the ratio of precipitated DNA in either wild-type or *haspin*[128] *Drosophila* larval brains relative to the control negative region. n = 3, means and s.e.m. are shown. C) Representative examples of homeotic transformations observed: second to first leg (L2-L1), third to first leg (L3-L1) and wing to haltere (W-H). A red circle marks sex combs that appear in the second and third leg of male flies. D) Frequencies of homeotic transformations in flies heterozygous for *Pc^1*, *Pc^3* or *Pc^15* mutations in either wild-type or homozygous *haspin*[128] mutant backgrounds. n = 3, over 20 individuals scored. E) Eye color of representative flies of a transgenic line P7[14] containing a homozygous PRE-*miniwhite* in a wild-type or *haspin*[128] mutant background (n = 3). F) Frequencies of homeotic transformations in flies heterozygous for *Pc^3* mutation in either wild-type or combinations of heterozygous mutant backgrounds for *haspin*[128] and cohesin complex components. n = 4, over 50 individuals scored. Statistical significance (*p<0.05, **p<0.01 and ***p<0.001) was determined by Student's *t*-test (A and B) or *z*-test (D and F).

Both in *Drosophila* and mammals, cohesin complexes have been shown to influence PcG-mediated gene silencing by supporting long-range interactions between Polycomb domains that are important for repression [46,47]. We have shown above that haspin is required for Pds5 binding to chromatin (Fig 2C) and modulates cohesin dynamics (S5 Fig). To analyze whether haspin and cohesin are functionally related in *Drosophila* we characterized homeotic transformations in flies with a combination of mutant alleles for *haspin* and cohesin complex components *Rad21*/*Vtd* and *Pds5*. While mutations of cohesin proteins alone do not significantly modify the *Pc³* extra sex combs phenotype (S9C Fig), the frequency of transformations increased when cohesin mutant alleles are in combination with *haspin¹²⁸* (Fig 5F). Indeed, flies containing both, *Pds5^e6* and *haspin¹²⁸* heterozygous mutant alleles, showed a strong enhancement of *Pc³* extra sex combs phenotype, since almost all the flies presented L2-L1 and L3-L1 transformations, with an average number of 7 sex combs in L2 and 4 in L3.

In summary, our data demonstrate that haspin is required for robust PcG-mediated regulation of gene expression and suggest that haspin and the cohesin complex, especially the cohesin-associated protein Pds5, cooperate to regulate homeotic gene silencing.

## Discussion

In this work we report that *Drosophila* haspin modulates key aspects of chromatin organization during interphase: insulator activity, heterochromatin-induced position-effect variegation, nuclear morphology and PRE-dependent pairing-sensitive silencing. Some of these aspects could be influenced by mitotic events and, therefore, be regulated by haspin functionality in chromosome organization during mitosis. However, our results also show that haspin phosphorylates histone H3 in interphase and associates with the cohesin complex mediating Pds5 binding to chromatin in interphase cells, strongly suggesting that this kinase modulates chromatin organization not only during mitosis but also in interphase.

Thr3 in histone H3 is located immediately adjacent to Lys4, which has been shown to be tri-methylated at active promoter sites [48,49]. TFIID binding to H3K4me3, which is involved in transcription machinery recruitment, is severely reduced in mitosis as a result of H3T3 phosphorylation [50]. On the other hand, in vitro studies have shown that tri-methylation of H3K4 reduces substrate recognition by haspin suggesting antagonism between H3K4 methylation and H3T3 phosphorylation [33]. In agreement with these studies, our results show that H3T3ph localizes at heterochromatin and lamin-enriched regions, while it does not colocalize with active RNA polymerase II. Even though, future investigations will be needed to decipher whether or not H3T3ph never colocalizes with H3K4me and active transcription, altogether these studies suggest antagonism between phosphorylated H3T3 and active transcription. Moreover, haspin is preferentially found at the nuclear lamina, like CP190 insulator protein and polycomb proteins (our results and previously reported studies [30,31]). While proteins that are associated with transcriptionally active chromatin are easily solubilized in subcellular fractionation assays, haspin, insulator and polycomb proteins are tightly bound to the nuclear matrix where they colocalize with laminas suggesting that nuclear organization of these proteins might contribute to their functionality.

We report here that haspin is a strong suppressor of position-effect variegation playing a role in heterochromatin organization. HP1 proteins, which are key components of heterochromatin, have been reported to promote haspin localization at mitotic centromeres to protect centromeric cohesion in mammals [51]. Whether HP1 proteins promote haspin localization at centromeric heterochromatin in interphase and whether phosphorylation of H3T3 is involved in heterochromatin organization remain to be determined.

We have also shown here that haspin is required for robust Polycomb-dependent homeotic gene silencing based on the following observations in the absence of haspin: i) derepression of *Abd-B* transcription, ii) reduction of Pc binding at several PRE-containing regulatory elements and iii) enhanced homeotic transformations in Pc mutant sensitized backgrounds. Moreover, depletion of haspin has a strong impact in pairing-sensitive silencing which involves long-range chromatin organization [11,52]. Our results also show that haspin cooperates with Pds5-cohesin to enhance Polycomb-dependent homeotic transformations. Thus, haspin might regulate homeotic gene silencing by directly affecting the binding of PcG proteins to chromatin or by affecting Pds5-cohesin dynamics modulating chromatin organization of Polycomb domains. Recent results point to an important role for cohesin complexes in the establishment and/or maintenance of Polycomb-repressed domains in mammalian cells but also to restrict their aggregation [46]. We have shown here that haspin mediates Pds5-binding to chromatin in interphase and modulates cohesin association with chromatin along the cell cycle. Pds5 proteins have both positive and negative effects on cohesin association with chromatin, they cooperate with Wapl in releasing cohesin from DNA but they are also required to maintain sister-chromatid cohesion in G2/M [30]. Pds5 interacts with Wapl, Dalmatian/Sororin, Eco/Eso acetyltransferase and haspin through the same conserved protein-protein module [21,53]. Wapl-Pds5 interaction has been shown to be counteracted by Eco and Sororin in S phase antagonizing Wapl's ability to dissociate cohesin from DNA [53,54]. On the other hand, haspin has been shown to phosphorylate Wapl [22] and to antagonize Wapl-Pds5 interaction to protect proper centromeric cohesion in mitosis [21,22,23]. Although relationship between haspin and the cohesin complex needs to be further characterized, our work point to an important role of haspin in the complex regulation of Pds5-cohesin dynamics along the entire cell cycle.

Pds5 proteins are also required for proper maintenance of heterochromatin [55] and participate in chromatin loop formation [56,57]. It has been suggested that they may be required for the boundary function of CTCF, since cells depleted of Pds5 proteins contain many fewer loops than control cells, which is similar to the effect of CTCF depletion [57]. On the other hand, it has been shown that chromatin becomes more compact after reducing levels of CTCF and Rad21 and the analysis of the molecular basis for this counter-intuitive behavior suggested that compaction could be the consequence of changes in chromatin loops [58]. Our data show that haspin is required for insulator activity, nuclear compaction, heterochromatin-induced position-effect variegation and PcG-mediated pairing-sensitive silencing strongly suggesting that haspin could be involved in the organization of the genome in chromatin domains and loops by modulating Pds5-cohesin association with chromatin.

It has been suggested that inhibition of haspin could have potent anti-tumoral effects with fewer adverse effects compared with other anti-cancer agents [59]. Our results show mitotic defects in *Drosophila* haspin mutants in agreement with previous reported studies in yeast and mammals [17,21]. However, haspin mutants have been reported not lethal in budding yeast [60] and in fission yeast [61] and our data show that they are also viable in *Drosophila*, even though life span and fertility are affected. Besides, our findings demonstrate that haspin is controlling genome organization of interphase cells raising concerns with respect to the use of haspin inhibitors as potent mitosis-specific anticancer drugs.

## Materials and methods

### *Drosophila* genetics and transgenic lines

Flies were raised in standard cornmeal yeast extract media at 25˚C except when indicated. $w^{1118}$ line was used as wild-type.

The mutagenesis screen was performed by mobilization of the P element from a P(Mae-UAS.6.11) line that carries the *yellow* (*y*) marker. The P element was mobilized from an X chromosome site in males and transpositions to the autosomes were recovered as y+ males. Insertions were subsequently screened for changes in *white* expression in enhancer-blocking assays with heterozygous flies of B7[27.1] transgenic line. This line carries an *attB* construct containing the *Fab7* boundary/insulator element (1.2 kb) between the *white* enhancer and the *mini-white* reporter gene (S1A Fig).

S1B and S1C Fig show the transgenic lines used in this study.

Constructs were inserted in the *attP51C* landing site via phiC31-mediated integration [62]. The RFP marker of the *attP* docking site and the *white* marker of the *attB* plasmid were eliminated via Cre recombinase-mediated excision, thus allowing enhancer-blocking assays.

For enhancer-blocking quantitative analyses, eye pigment of 20 heads was extracted with 30% acid-ethanol (pH 2) according to [63] and $OD_{480}$ was determined in a Nanodrop 1000/3.7.

*CP190[RNAi]* and *CTCF[RNAi]* lines from Vienna *Drosophila* RNAi Center (#35078 and #30713 respectively). *haspin[RNAi]* lines from BDSC (B-35276 and B-57787). *GFP[RNAi]* transgenic line was generated according to standard procedures. *CP190[H31-2]* [64]. *Pds5[e6]*, *Vtd[ex3]*, *Vtd[ex3]*,*tub-Rad21-myc*, *In(1)w[m4]*, GAL4 lines (*da-GAL4*, *nub-GAL4*, *69B-GAL4* and *Act5C-GAL4*) and Polycomb mutations (*Pc[1]*, *Pc[3]* and *Pc[15]*) were obtained from BDSC. *da-GAL4* line does not contain the *mini-white* reporter gene to allow enhancer-blocking assays. *69B-GAL4* was used to drive moderate expression in larval salivary glands [65].

*Haspin* cDNA (LD07633 from *Drosophila* Genomics Resource Center) was cloned to be expressed under the control of either 3 kb of *haspin* 5' genomic sequences (*haspin[PROM]*) or UAS sequences (tagged HA versions of the protein either wild-type, *UAS-haspin-HA*, or with point mutations in the kinase domain, *UAS-haspin[H420A]-HA* and *UAS-haspin[K282M]-HA*). Details of the constructs are available upon request. *UAS-Pds5-HA-Flag* construct was obtained from DGRC (UFO12474). Transgenic lines with these constructs were obtained via phiC31-mediated integration [62].

Longevity assays were performed with 80 newly hatched either females or males housed at 20 flies/vial and transferred to fresh vials every 5 days. For the fertility test, newly hatched females or males were mated with 3 *w* virgin individuals of the opposite sex and at least ten crosses for control and *haspin[128]* mutant flies were set up simultaneously. Flies were transferred into fresh vials with 3 new *w* flies of the opposite sex every 5 days until they were 20-day old. The progenies of each cross were counted excluding vials that did not contain all four flies alive at the end of each 5-day mating period.

## Antibodies

Rat αhaspin polyclonal antibodies were raised against bacterially expressed recombinant protein containing amino acids 1–316 of haspin and were validated in western blot, immunostaining and immunoprecipitation assays giving specific signal only in IP assays (IP 3μl). Rabbit αCP190 is described in [66] (WB 1:5,000) and rat αCenp-C is a gift from F. Azorin (IF 1:200). Commercially available antibodies used were as follows: rat αHA (Roche 1867423, WB 1:500, IF 1:100), rabbit αH3 (Cell Signaling 9715, WB 1:5000), mouse αlaminDm0 (Developmental Studies Hybridoma Bank ADL67, WB 1:2,500, IF 1:500), rabbit αH3T3ph (Millipore 07–424, IF 1:100, Millipore 04–746, ChIP 3μl), rabbit αPc (Santa Cruz Biotech, WB 1:500, ChIP 3μl), mouse αmyc (Millipore 05–724, WB 1:1000).

## Immunostaining experiments

Immunostaining of larval salivary glands, polytene chromosomes and neuroblast squashes were performed as described elsewhere [66,67,68]. For visualization, slides were mounted in

Mowiol (Calbiochem-Novabiochem) containing 0.2ng/ml DAPI (Sigma) and visualized in a confocal Leica TCS SP2-AOBS microscope. Images were acquired and processed using ImageJ (http://imagej.nih.gov/ij/) and Adobe Photoshop software. At least three independent biological replicates were performed.

## Quantification of immunofluorescence images

Mean grey areas (nuclei) or intensities (polytene chromosomes) were calculated using ImageJ on thresholded images at DAPI masked regions of interest running Analyze particles plugin on the FeatureJ.

## Coimmunoprecipitation experiments

Assays were performed with extracts prepared from *Drosophila* salivary glands. Cells were lysed with 0.5% NP40, 300mM NaCl, 50mM Tris pH8, 5mM EDTA and protease inhibitors. Incubation with αhaspin, αFlag or no antibodies was performed at 4˚C. After incubation with Protein A/G Agarose (Santa Cruz Biotech), beads were pelleted by centrifugation, washed and analysed by Western-blot. At least two independent biological replicates were performed.

## Biochemical fractionation

3–21 hours old embryos, collected in juice agar plates and dechorionated, were subjected to loose dounce in ENB buffer (0.3M sucrose, 10mM Tris-HCl pH 8, 60mM KCl, 15mM NaCl, 1mM EDTA, 0.5mM EGTA, 0.5mM spermidine, 0.15mM spermine, 0.1mM PMSF and Protease Inhibitor Cocktail). Centrifugation was carried out for 5 min at 2,300g resulting in supernatant (cytoplasm + some nuclear soluble fraction) and pellet (nuclei). The pellet was incubated in CSK buffer (0.3M sucrose, 10mM Hepes pH 7.9, 250mM NaCl, 3mM MgCl$_2$, 0.05mM CaCl$_2$, 0.5mM DTT, 0.1mM PMSF and Protease Inhibitor Cocktail) for 5 min at 0˚C and centrifuged to obtain the nuclear soluble fraction. Then, the pellet was incubated in CSK buffer containing 10U DNAseI for 15 min at 37˚C and NaCl was added to 2M final concentration. After 5 min at 0˚C samples were centrifuged to obtain the high-salt chromatin fraction and the remaining pellet containing the nuclear matrix fraction was solubilized in 8M Urea, 0.1M NaH$_2$PO$_4$, 10mM Tris-HCl pH8. To be able to compare protein amounts in the different fractions equal buffer volumes were used for all fractionation steps except for the first one, which was carried out in a large volume (2x) to purify properly nuclei. Then, double volume of this fraction with respect to the others were subjected to SDS-PAGE and immunoblotted with the indicated antibodies.

## Chromatin purification

Nuclei from 0–4 and 20–24 hours embryos were prepared as described in the biochemical fractionation procedure. Nuclei were crosslinked for 10 min in PBS with 1% formaldehyde. 0.125M glycine (final concentration) was added to stop reaction and nuclei were pelleted by centrifugation at 2,300g for 5 min. Nuclei were washed sequentially for 10 min each in WA (0.25% Triton X-100, 10mM Tris-HCl pH8, 10mM EDTA, 0.5mM EGTA), WB (0.2M NaCl, 10mM Tris-HCl pH8, 1mM EDTA, 0.5mM EGTA) and WC (1% SDS, 10mM Tris-HCl pH8, 1mM EDTA). To revers crosslinks pellets were resuspended in 25mM Tris-HCl pH6.8, 4.4% glycerol, 1% SDS, 5% 2-mercaptoethanol, 0.005% bromophenol blue and incubated for 20 min at 95˚C. Samples were subjected to SDS-PAGE and immunoblotted with the indicated antibodies.

## Quantitative RT-qPCR

Total RNAs were extracted with TRIzol Reagent (Life Technologies), purified with RNeasy Mini Kit (QIAGEN) and treated with DNAseI (QIAGEN). cDNAs were prepared from 0.8 μg of RNA using the Transcriptor First Strand cDNA Synthesis Kit (Roche) and oligo-dT primers.–RT controls were included in qPCR reactions to discard genomic DNA contamination. qPCR was performed on Roche LightCycler 480 System (Roche) using LightCycler 480 SYBER Green I Master (Roche). Primers used are listed in S1 Table. Three independent biological replicates for each genotype were performed.

## DNA content analysis

DNA from S2 cells and 20 larval salivary glands either WT or *haspin*[86] was extracted by standard methods. DNA content was calculated by real-time qPCR. The logarithm values of 3 quantities of DNA standard (S2 cells) in 10-fold dilutions and the corresponding cycle numbers (Ct value) of actin gene amplification were used to perform linear regression. This standard was used to calculate the relative DNA content in WT and *haspin*[86] mutant salivary glands (see S6C Fig in supporting information spreadsheet form).

## ChIP experiments

For ChIP, chromatin from S2 cells was fixed in 1.8% formaldehyde for 10 min at room temperature. Cells were sonicated in TE containing 0.1% SDS in a Diagenode Bioruptor. *Drosophila* brains of third-instar larvae were fixed in 1% formaldehyde for 15 min at room temperature. Brains were resuspended in 10mM Tris-HCl pH 8, 10mM NaCl, 0.2% NP-40 and manually homogenized. After centrifugation pellet was resuspended in lysis buffer (50mM Tris-HCl pH8, 10mM EDTA and 1% SDS) and incubated for 20 min at room temperature. A 0.5x volume of dilution buffer (1.1% Triton X100, 16.7mM Tris-HCl pH8, 2mM EDTA, 167mM NaCl) was added and chromatin was sonicated in a Branson sonifier. Samples from S2 cells and brains were immunoprecipitated in RIPA buffer (140mM NaCl, 10mM Tris-HCl pH8, 1mM EDTA, 1% Triton X100, 0.1% SDS, 0.1% sodium deoxycholate) with Protein A Sepharose. For ChIP-qPCR, three independent biological replicates were analyzed. Primers used are listed in S1 Table. ChIP-seq libraries for Illumina sequencing were constructed following manufacture's protocols.

## Supporting information

**S1 Fig. Transgenic lines used in enhancer-blocking assays.** A) Diagram of the reporter used in enhancer-blocking assays. **E** corresponds to *White* enhancer sequences (X: 2798339–2796777) that contain the eye enhancer [70]. **RE** indicates the different regulatory elements and *mini-white* is the reporter gene. B) Scheme of *Abd-B* genomic region with the regulatory elements used in this study. **B** indicates boundary/insulator element and **P** indicates PRE. C) Locations of the fragments corresponding to the regulatory elements in the different transgenic lines are indicated.
(TIF)

**S2 Fig. Phenotypes of *Drosophila* haspin mutant alleles.** A) Scheme of *haspin* genomic organization in *haspin*[86] and *haspin*[128] mutant lines. Location of the P element is indicated by a triangle and deleted sequences in line *haspin*[128] are indicated in grey. B) *Haspin* transcriptional levels normalized to *Actin5C* as fold changes relative to control in *haspin*[86] and *haspin*[128] mutant larvae. n = 3, means and s.d. are shown. C) Survival curves for wild-type and *haspin*[128] adult flies. n≥5, means and s.d. are shown. D) Fertility test: progenies of wild-type and

*haspin^128* flies of the indicated ages were counted and plotted (n≥20). In these tests *haspin^128* females did not survive more than 15–20 days. Statistical significance (*p<0.05, **p<0.01 and ***p<0.001) was determined by Student's *t*-test (panels in B and C) or Wilcoxon test (panels in D).
(TIF)

**S3 Fig. Mitotic defects in haspin mutant backgrounds.** A) Immunolocalization at higher magnification of H3T3ph (green) and Cenp-C (red) on chromosome spreads prepared from wild-type and *haspin^128 Drosophila* larval brains arrested in mitosis with colcemid. DNA is stained with DAPI (blue). Scale bars are 2 μm. B) Box plot showing quantification of inter-kinetochore distances in wild-type and *haspin^128* chromosome spreads. The inter-kinetochore distance was measured using the centromere marker Cenp-C in over 15 chromosomes. Distance was determined by drawing lines which length was calculated using the imageJ software. ***p<0.001 as determined by Wilcoxon test.
(TIF)

**S4 Fig. Haspin is required for Pds5-chromatin interaction.** A) Western blot analysis using αHA of salivary gland extracts from larvae that express haspin-HA under the control of *Actin5C* promoter that were subjected to immunoprecipitation with αhaspin. Input corresponds to 10% of the immunoprecipitated material. B) Western blot analysis using αFlag of salivary gland extracts from larvae that express Pds5-HA-Flag under the control of *nubbin* promoter that were subjected to immunoprecipitation with αhaspin or αFlag. Input corresponds to 10% of the immunoprecipitated material. C) *Pds5* transcriptional levels normalized to *Actin5C* as fold changes relative to control in *haspin^128* mutant *Drosophila* third-instar larvae (L) and salivary glands of third-instar larvae (SG). n = 3, means and s.d. are shown. D) *Drosophila* salivary glands of third-instar larvae that express Pds5-HA under the control of *nubbin* promoter in wild-type (upper panels) or haspin RNAi background (lower panels) immunostained with antibodies against HA. DNA is stained with DAPI.
(TIF)

**S5 Fig. Haspin modulates cohesin-chromatin interactions.** A) Western blot analysis using αmyc of salivary gland extracts from larvae that express ubiquitously Rad21-myc in a Rad21 mutant background that were subjected to immunoprecipitation with αhaspin. Input corresponds to 10% of the immunoprecipitated material. B) Representative polytene chromosome spreads from salivary glands of third-instar larvae that express Rad21-myc, under the control of *tubulin* promoter in a Rad21 mutant background (*Vtd^{ex3}*), in control (upper panels) or *haspin^128* mutant background (lower panels) immunostained with antibodies against myc (red) and CP190 (green). C) *Rad21* transcriptional levels normalized to *Actin5C* as fold changes relative to control in *haspin^128* mutant *Drosophila* third-instar larvae (L) and salivary glands of third-instar larvae (SG). n = 3, means and s.d. are shown. D) Western blot analysis using αmyc (upper row) of chromatin extracts from *Drosophila* embryos from 0–4 h (left panel) and 20–24 h (right panel) after egg laying that express Rad21-myc in a Rad21 mutant background (*Vtd^{ex3}*) in control or *haspin^128* mutant backgrounds. Antibodies to H3 were used for the loading control (bottom row). E) Rad21-myc protein levels normalized to H3 in chromatin extracts of control or *haspin^128* mutant *Drosophila* embryos. Error bars are s.d. of three independent biological replicates. Differences in chromatin associated Rad21 are statically significant (*p<0.05 and **p<0.01 as determined by Student's *t*-test).
(TIF)

**S6 Fig. Haspin Kinase is localized at the nuclear matrix and modulates nuclear architecture.** A) Biochemical fractionation of *Drosophila* embryos that express haspin-HA under the

control of *Actin5C* promoter. Aliquots of cytoplasm + nuclear soluble (lane 1), nuclear soluble (lane 2), chromatin (lane 3) and nuclear matrix (lane 4) fractions were subjected to SDS-PAGE and immunoblotted with the indicated antibodies. B) Immunostaining of *Drosophila* salivary glands with DAPI in larvae of the indicated genotypes. Scale bars represent 50 μm. C) Relative DNA content in wild-type and *haspin*[86] *Drosophila* larval salivary glands. n = 3, means and s.d. are shown. D) Western blot analysis of larval salivary gland protein extracts of control and overexpression of either wild-type protein (nub > *haspin-HA*) or mutated proteins in the kinase domain (nub > *haspin*[H420A]*HA* and nub > *haspin*[K282M]*HA*) using antibodies to HA (upper row). Antibodies to H3 were used for the loading control (bottom row).
(TIF)

**S7 Fig. Genomic distribution of H3T3ph.** A) Proportion of H3T3ph peaks that overlap mod-ENCODE HP1a enriched regions. Association was analyzed using overlap permutation tests with the *overlapPermTest* function from the *regioneR* package version 1.14.0 using 5000 permutations and default options. The z-score numerical measurement indicates the strength of the association. B) Proportion of H3T3ph peaks in the 9 chromatin states characterized by [34]. State 1 (red) active promoters and transcription start sites; state 2 (yellow) transcript elongation; states 3 and 4 (light and bright green) regulatory regions; state 5 (green-blue) active male X chromosome; state 6 (light blue) PcG regions; state 7 (dark blue) centromeric heterochromatin and chromosome 4; state 8 (purple) other heterochromatin; state 9 (pink) other silent domains. z-score in permutation tests is indicated below (blue and red indicate negative and positive values respectively). C) Proportion of euchromatic H3T3ph peaks, which were defined as those not overlapping with chromatin state 7, in chromatin states characterized by [34].
(TIF)

**S8 Fig. Genomic distribution of H3T3ph at euchromatin.** A) ChIP-seq data for H3T3ph over a region of 800 kb in chromosome 3 that contains the Antennapedia complex. Binding profiles of CP190 and Ibf2 [66], Pol II S5ph and H3K27me3 (modENCODE data) and lamin Dm0 [69] are depicted. High, low and no signal for H3T3ph, H3K27me3 and lamin Dm0 are indicated below by black, grey and white bars respectively. B) Whole genome colocalization of H3T3ph with Pol II S2ph/S5ph.
(TIF)

**S9 Fig. Haspin mutant backgrounds enhance Pc-associated phenotypes.** A) Frequencies of male L2-L1 and female W-H homeotic transformations in a *Pc*[3] heterozygous mutant background either expressing or not *haspin*[RNAi] under the control of the Actin5C promoter at 25˚C. Increasing temperature to 29˚C caused male lethality and only female W-H transformations were scored. n = 3, over 50 and 8 individuals scored at 25˚C and 29˚C respectively. ***p<0.001 as determined by *z*-test. B) The percentage of homozygous F7[5F24] transgenic females at 25˚C showing normal wings (0) and wing blade destruction in one (1) or both (2) wings in wild-type and haspin mutant flies is presented. n indicates number of females scored. C) Frequencies of homeotic transformations in flies heterozygous for *Pc*[3] mutation in either wild-type or heterozygous mutant backgrounds for cohesin complex components. n = 3, over 50 individuals scored. No significant differences as determined by *z*-test.
(TIF)

**S1 Table. Primers used in RT-qPCR and ChIP-qPCR.**
(DOCX)

**S1 Text. Bioinformatics analyses of ChIP-Seq data.**
(DOCX)

**S1 Data. Spreadsheet with numerical data used to generate graphs in all Figures.**
(XLSX)

## Acknowledgments

We thank Josep Casacuberta, Elena Casacuberta, Joan Roig, Jordi Casanova, F. Azorin and all members of his laboratory for helpful discussions and comments. We are grateful to F. Azorin and R. Paro for antibodies and fly stocks. We are thankful to Bloomington Drosophila Stock Center, Vienna Drosophila RNAi Center, Drosophila Genomics Resource Center and Developmental Studies Hybridoma Bank for providing fly stocks, clones and antibodies. We also thank Alicia Vera for technical assistance.

## Author Contributions

**Conceptualization:** Ujué Fresán, Maria A. Rodríguez-Sánchez, M. Lluisa Espinàs.

**Formal analysis:** Ujué Fresán, Maria A. Rodríguez-Sánchez, Oscar Reina, M. Lluisa Espinàs.

**Funding acquisition:** M. Lluisa Espinàs.

**Investigation:** Ujué Fresán, Maria A. Rodríguez-Sánchez, M. Lluisa Espinàs.

**Supervision:** Victor G. Corces, M. Lluisa Espinàs.

**Writing – original draft:** Oscar Reina, M. Lluisa Espinàs.

**Writing – review & editing:** Ujué Fresán, Maria A. Rodríguez-Sánchez, Oscar Reina, Victor G. Corces, M. Lluisa Espinàs.

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
