## [Decision Letter · Decision Letter 0]

17 Dec 2019

Dear Dr Espinás,

Thank you very much for submitting your Research Article entitled 'Haspin kinase modulates nuclear architecture and Polycomb-dependent gene silencing' to PLOS Genetics. Your manuscript was fully evaluated at the editorial level and by independent peer reviewers. The reviewers appreciated the attention to an important problem, but raised some substantial concerns about the current manuscript. Based on the reviews, we will not be able to accept this version of the manuscript, but we would be willing to review again a much-revised version. We cannot, of course, promise publication at that time.

If you decide to revise the manuscript for further consideration at PLOS Genetics, please aim to resubmit within the next 60 days, unless it will take extra time to address the concerns of the reviewers, in which case we would appreciate an expected resubmission date by email to plosgenetics@plos.org.

[LINK]

We are sorry that we cannot be more positive about your manuscript at this stage. Please do not hesitate to contact us if you have any concerns or questions.

Yours sincerely,

Kami Ahmad

Guest Editor

PLOS Genetics

Wendy Bickmore

Section Editor: Epigenetics

PLOS Genetics

Reviewer's Responses to Questions

**Comments to the Authors:**

Reviewer #1: Haspin is a conserved kinase from yeast to human, known to play a role in centromere cohesion during mitosis. It has been mostly studied in yeast and human. In this article the authors analyze Haspin in Drosophila. While they confirm a role of the kinase during mitosis, the authors also document activity during interphase where Haspin is shown to be associated with euchromatin where it is involved in Position-Effect-Variegation and Polycomb-mediated silencing. Given the conservation of the kinase through evolution, and its implication in well-tried Drosophila genetic system used to analyze processes regulated by chromatin in general, I find these new informations of interest for the general readership of PLoS Genetics. I list below a number of comments that could improve the quality of the paper .

Sometimes, Suppression of enhancer blocking activity of insulators can be sensitive to the background. The authors describe a rescue construct for Haspin (Haspin-prom). If they have the rescue construct recombined on a chromosome carrying Haspin (86) it should be a simple cross to document reversion of the Suppressor effect. I usually hate to ask for additional experiments and the authors showed that knocking down Haspin by RNAi also leads to the suppression of Fab-7-mediated enhancer blocking activity. So my suggestion is not mandatory, but if the rescue experiment could be added, it would be perfect.

At page 5 the authors should inform right away the readers that flies apparently knocked down for haspin are viable. Are the homozygotes fertile? What is known about maternal contribution? Looking at Flybase (which by the way mentions the existence of 5 uncharacterized alleles), there is a moderate presence of the mRNA in 0-2 hours embryos. The authors mention semi fertility of the female. Do these eggs develop through adulthood?

I find difficult to monitor the increase in the distance between centromeres of mitotic chromosomes in metaphase in the Haspin mutant background (Fig.2). Somehow the green H3T3ph signal in between the cenp-C spots blur the aspect. It would be perhaps useful to display a red channel only panel for the WT chromosome.

In the middle of page 7, the authors describe the generation of an antibody against Haspin protein. Have they tried staining of salivary gland? It would be interesting to correlate it with the staining detecting phosphorytaled H3T3. By the way did the authors checked if the BX-C is decorated by H3T3ph? I was not able to recognize 3R on the spread displayed in FigS4, but the quality of their squash is fine and a direct observation under the microscope would perhaps allow to find the BX-C. I also wonder if it is possible to correlate the H3T3ph chipseq data with the staining on salivary gland.

Bottom of page 7 the authors mention no apparent change of Rad21 binding to salivary gland giant chromosome in the absence of Haspin. But it seems to me that there is a decrease in intensity and in the number of sites decorated by Rad21 in haspin mutant background. In general, I also find that the quality of the squashes generated in Haspin mutant background is lower, with thinner and less-well spread chromosomes. Is that a general trend or a coincidence? The authors should check if Rad21 expression is not decreased in haspin mutant background.

I am not entirely sure of what to do with the observation that Abd-B expression increases in the haspin(86) mutant background. If this increase originates from a weakening of the Pc-G mediated repression, one should also observe ectopic activation leading to slight gain-of-function phenotype such as the appearance of pigmentation on the 4th male tergite. I imagine that the authors would have noticed it. So the lack of extra-pigmentation is puzzling to me. Abd-B is hapo-insufficient (this is why it is written with an uppercase A). Thus Abd-B/+ males have often a few bristles on the 6th sternite as well as slightly extended genitalia. I wonder if this haplo-insufficiency can be compensated by introducing the haspin mutation? This could be also tested for the haplo-insufficient phenotype of Ubx mutations (Ubx/+ have slightly increase halters).

Regarding the remark in the middle of page 13 about the difference in extra sex comb formation during the 1st three generations, it is not necessary to invoke trans-generational inheritance. It is indeed a well-known fact that Pc-G stocks accumulate modifier mutation that suppress their haplol-insufficient phenotype. By simply out-crossing, these modifiers loci are diluted out. The authors should have observed the same phenomenon occurring even with the WT chromosome 2 control.

Finally, as a dosophilist, I am sensitive to the naming of mutations that most of the time relate to the process involved. As a matter of facts many geneticists were/are very creative in naming their favorite mutations. I am therefore curious to know why this kinase is named “Haspin”. I understand that in this case there will be no relationship between the name of the mutation and the phenotype. Given the fact that the homozygotes do not show an apparent phenotype, I suggest writing the genotype with a lowercase 1st letter “haspin”.

Reviewer #2: This manuscript aims to detail a novel role of the kinase Haspin in genome organisation. The authors confirm the previously described role of Haspin in protecting centromeric cohesion in mitosis, but they additionally suggest a new role during interphase. They show that Haspin, previously reported to be a mitosis specific kinase, is capable of phosphorylating its only well-defined target, H3T3, in Drosophila polytene chromosomes in interphase. Using mutant flies, they show that insulator activity is disrupted in the absence of Haspin and suggest that long range genome interactions may be affected due to a lack of chromosomal PDS5 binding in these mutants. The authors also suggest a role of Haspin in PEV and polycomb dependent gene silencing. There are several very interesting observations in this study that, if substantiated, would significantly broaden our understanding of where Haspin functions. However, there are key questions regarding the properties of the H3T3ph antibodies used, and the rigour of analysis of ChIP-seq data. In addition, the mechanisms underlying the phenotypes are unclear. It remains a question whether a new interphase function of Haspin has been revealed, or whether many of these observations can be explained through Haspin’s known function in mitosis.

Major points:

1. For readers who are not fly specialists, it would be useful to describe the eye-colour change assay more clearly. In addition, it is unclear how strong the observed phenotype is compared to other regulators of this system. Can controls without the insulator element or with a knockdown of a protein with known insulator activity be carried out?

2. The finding of H3T3ph during interphase is novel and exciting. However, all the data to support this are in polytene chromosomes while other aspects of this study are in other cell types. Can H3T3ph be detected in interphase in other cells types or is this a polytene chromosome specific phenomenon (nuclear H3T3ph staining should be simple to observe)? In addition, is H3T3ph on polytene chromosomes enriched at the centromeres as seen in mitosis?

3. The evidence that Haspin really functions in interphase (other than H3T3ph on polytene chromosomes is weak). The experiment in Figure S4E is not compelling because it is very indirect and in fact shows that the extent of Rad21 localisation actually correlates with the %mitotic cells in two populations.

4. The IF results in Figure 2 should be quantified to show the effects are consistent across many cells, and to give some idea of the extent of protein loss from chromosomes. Does PDS5 also disappear from chromosomes in Haspin mutant cells?

5. PDS5 proteins have been shown to bind to SCC1 directly. Figure 2C shows that, in the absence of haspin, PDS5 cannot bind to chromatin yet the absence of haspin does not prevent the binding of the majority of the core cohesin complex. This is interesting, but contrary to the popular model that PDS5 localises to chromatin through cohesin and Haspin is localised through PDS5. This raises the question of whether the loss of PDS5 is due to Haspin acting as a chromosome receptor for PDS5, or whether it is an indirect effect resulting from Haspin’s function in regulating cohesion. Can this be addressed For example, does haspin co-IP with PDS5? (It might also be interesting, but not essential, to know if the chromatin fraction in a haspin knockdown no longer contains PDS5 and it is instead found in the soluble fraction).

6. Biochemical fractionation (figure S5A) shows no Haspin in the chromatin fraction, which is somewhat surprising and perhaps should be explained/commented on.

7. It would be useful to know the extent to which Haspin-HA is overexpressed. It is surprising that Haspin-HA overexpression returns nuclear size to normal in the Haspin mutant (Fig 3C) when a single additional copy of the Haspin gene is sufficient to increase nuclear size (Fig 3B).

8. The antibody used in the H3T3ph ChIP-seq experiments has not been sufficiently validated for use in ChIP-seq. It is common that adjacent modifications can prevent histone antibody binding. In this case, H3K4me3 is of particular concern and the authors make several conclusions regarding K4me3, TFIID and RNA pol II. Without validation of the antibody’s ability to bind to H3T3ph adjacent to K4me3, these claims are not supported by the presented data. We suggest that the authors carry out an ELISA or similar assay to address this issue (in fact our own data indicate that JY325 is strongly affected by H3K4 methylation).

9. Visual inspection of ChIP-seq data is insufficient to draw conclusions about the colocalization of H3T3ph and other factors. Bioinformatics analysis showing the statistical significance of enrichment of multiple factors in various chromatin states or extent of peak overlap should be shown to validate this.

10. The authors typically equate Haspin action with H3T3 phosphorylation, but the difference should be made clear. Haspin has non-kinase dependent roles, and it also should be acknowledged that Haspin is likely to have substrates other than H3T3ph (eg Wapl).

11. Throughout, the number of repeats (n=?) should be provided in figure legends.

Minor Points:

1. There are several grammatical errors and typos. Examples include use of the word ‘being’ mid-sentence (abstract and several other sentences); commas after the word “both” are in all cases misplaced and confusing; hapin instead of haspin (p. 9), different not diferent (p.5).

2. Figure 1A legend does not match labels on the figure (B7 or F7 transgenic line)

3. Generally, the figure legends should provide a better description of the abbreviations used in the figures. Particular examples include figure 1 C and D

4. Molecular weight markers should be shown on the western.

5. In figure 3, details on how nuclear size is defined and quantified should be expanded. The same applies to figure S5C for relative DNA content.

6. The use of chromatin compaction (page 8) is misleading when referring to a change in nucleus size as the data does not address whether the chromatin itself changes.

7. In figure S6A, H3T3ph is not found in state 8 defined as “other heterochromatin” but is largely found in state 7 which is defined as “centromeric heterochromatin”. The authors should discuss these states in more detail and provide a suggestion as to why they believe this is not just the known mitotic centromeric H3T3ph signal. In addition, the authors claim a colocalization with HP1a or lamin – are these also enriched in these states?

8. P. 3. There is very good evidence that H3S10ph is not required for chromosome condensation, particularly in mammals. This sentence should be rephrased.

9. P.4. Haspin has not been shown to regulate cohesion in fission yeast.

**Have all data underlying the figures and results presented in the manuscript been provided?**

Reviewer #1: Yes

Reviewer #2: No: I did not see a spreadsheet for underlying numerical data.

PLOS authors have the option to publish the peer review history of their article (what does this mean?). If published, this will include your full peer review and any attached files.

Reviewer #1: Yes: Francois Karch

Reviewer #2: No

---

## [Decision Letter · Decision Letter 1]

29 Jun 2020

Dear Dr Espinás,

We are pleased to inform you that your manuscript entitled "Haspin kinase modulates nuclear architecture and Polycomb-dependent gene silencing" has been editorially accepted for publication in PLOS Genetics. Congratulations!

Yours sincerely,

Kami Ahmad

Guest Editor

PLOS Genetics

Wendy Bickmore

Section Editor: Epigenetics

PLOS Genetics

Comments from the reviewers (if applicable):

Reviewer's Responses to Questions

**Comments to the Authors:**

Reviewer #1: The authors have given satisfactory answers to my questions. Regarding the experiments displayed in Figure 5A, it does not really make sense to document Abd-B expression from ovaries, a tissue in which the expression of this gene is not well documented. Authors MUST indicate at which stage they have dissected out the brains. I suspect larvae. In the unlikely case they dissected them out of adult,, they should then indicate if it is nerve chord or brain.

Reviewer #2: The manuscript is considerably improved since the first submission, particularly by the inclusion of some statistical testing of the ChIP-seq results. It provides novel and very interesting information on the function of Haspin in flies that is highly likely to stimulate work in other organisms.

I think the authors could be given the opportunity to address some minor points prior to publication:

1. In the introduction it is stated that H3T3ph has been shown to have a role in cohesion regulation, but the cited reference only shows a role for Haspin, not for H3T3ph.

2. In my view, Fig S4C would be better included in the relevant main figure.

3. The decline in Pds5 protein levels when Haspin is reduced is shown only by IF. This seems less than ideal, because delocalisation of haspin could lead to it becoming more extractable in IF conditions. Furthermore, if total Pds5 protein levels do fall substantially, how can the authors be sure that Haspin depletion really prevents Pds5 binding chromatin? If there is no Pds5 protein, then it won't be seen on chromatin. Because of these points, I think the relevant Results and Discussion sections need to be more carefully worded.

4. I think the authors perhaps should be more conscious of the distinction between observing phenotypes in interphase cells, and whether the cause of these changes has to be due to events directly regulated by Haspin in interphase. It seems conceivable to me that Haspin might have a function only in mitosis, but these have knock-on effects on PEV etc. For example, if Haspin is involved in the appropriate maintenance of heterochromatin or insulator function through mitosis, then similar phenotypes to those observed may emerge in interphase. This goes back to my original point about H3T3ph not being directly observed in interphase cells except on polytene chromosomes. Of course the authors' own very interesting model is also consistent with the data, but the published work on Haspin to date is very much focused on mitosis.

5. Although the paper is generally well referenced, it is a bit surprising that the original study showing that Haspin is an H3T3 kinase (Dai et al, Genes Dev 2005) is not mentioned.

**Have all data underlying the figures and results presented in the manuscript been provided?**

Reviewer #1: Yes

Reviewer #2: Yes

PLOS authors have the option to publish the peer review history of their article (what does this mean?). If published, this will include your full peer review and any attached files.

Reviewer #1: **Yes: **François Karch

Reviewer #2: No

**Data Deposition**

http://datadryad.org/submit?journalID=pgenetics&manu=PGENETICS-D-19-01708R1

**Press Queries**

---

## [Editor Report · Acceptance letter]

28 Jul 2020

PGENETICS-D-19-01708R1 

Haspin kinase modulates nuclear architecture and Polycomb-dependent gene silencing 

Dear Dr Espinás, 

We are pleased to inform you that your manuscript entitled "Haspin kinase modulates nuclear architecture and Polycomb-dependent gene silencing" has been formally accepted for publication in PLOS Genetics! Your manuscript is now with our production department and you will be notified of the publication date in due course.

With kind regards,

Matt Lyles

PLOS Genetics

On behalf of:
